# Plagues of Desert Locusts: Very Low Invasion Risk to China

**DOI:** 10.3390/insects11090628

**Published:** 2020-09-11

**Authors:** Yun-Ping Wang, Ming-Fei Wu, Pei-Jiong Lin, Yao Wang, Ai-Dong Chen, Yu-Ying Jiang, Bao-Ping Zhai, Jason W. Chapman, Gao Hu

**Affiliations:** 1College of Plant Protection, Nanjing Agricultural University, Nanjing 210095, China; 2017102066@njau.edu.cn (Y.-P.W.); 2017102067@njau.edu.cn (M.-F.W.); 2018102071@njau.edu.cn (P.-J.L.); 2018102070@njau.edu.cn (Y.W.); bpzhai@njau.edu.cn (B.-P.Z.); j.chapman2@exeter.ac.uk (J.W.C.); 2Agricultural Environment and Resources Institute, Yunnan Academy of Agricultural Sciences, Kunming 650205, China; shenad68@163.com; 3China National Agro-Tech Extension and Service Center, Beijing 100125, China; jiangyuying@agri.gov.cn; 4Centre of Ecology and Conservation, and Environment and Sustainability Institute, University of Exeter, Penryn, Cornwall TR10 9FE, UK

**Keywords:** *Schistocerca gregaria*, insect migration, trajectory simulation, locust upsurge, invasion risk

## Abstract

**Simple Summary:**

During the latest upsurge of the desert locust, swarms in northern Pakistan and India approached the Chinese border areas. In view of the recent windborne movement into China of another very serious migratory pest, the fall armyworm moth, we investigated the risk of the desert locust invading China. Our analysis shows that China is not likely to experience invasions of desert locust swarms (except, possibly, very marginal incursions on the Nepal border) and there was no likelihood that desert locust would become established in China over the next few decades.

**Abstract:**

Recently, the most serious upsurge of the desert locust (*Schistocerca gregaria*) in the last 25 years is spreading across eastern Africa and southwestern Asia. Parts of the desert locust ‘invasion area’, namely the northern border areas of Pakistan and India, are very close to China, and whether locust swarms will invade China is of wide concern. To answer this question, we identified areas of potentially suitable habitat for the desert locust within China based on historical precipitation and temperature data, and found that parts of Xinjiang and Inner Mongolia provinces could provide ephemeral habitat in summer, but these places are remote from any other desert locust breeding areas. New generation adults of the desert locust in Pakistan and India present since April led to swarms spreading into the Indo-Pakistan border region in June, and so we examined historical wind data for this period. Our results showed that winds at the altitude of locust swarm flight blew eastward during April–June, but the wind speeds were quite slow and would not facilitate desert locust eastward migration over large distances. Simulated trajectories of desert locust swarms undertaking 10-day migrations mostly ended within India. The most easterly point of these trajectories just reached eastern India, and this is very close to the eastern border of the invasion area of desert locusts described in previous studies. Overall, the risk that the desert locust will invade China is very low.

## 1. Introduction

The desert locust, *Schistocerca gregaria*, is one of the most devastating migratory pests in the world [1,2,3,4]. It is highly mobile and feeds on large quantities of any kind of green vegetation, including crops, pasture, and fodder [2,3], and even a moderate swarm measuring 10 km^2^ would eat some 1000 tons of vegetation daily. Recently, the most serious upsurge of desert locusts since the last serious outbreak in 1994–1995 is spreading across eastern Africa and southwestern Asia. When this study began in late February 2020, more than 280,000 ha in 13 countries were infected [5,6], and the latest figures published by FAO (late May 2020) showed that the infected area had increased up to 332,000 ha. [7]. Among the ‘Invasion areas’ defined from earlier studies [1], the northern borders of Pakistan and India are very close to China [5,7]. China is struggling to control fall armyworm (*Spodoptera frugiperda*), a migratory pest that invaded China via India and Myanmar last year [8], and consequently, whether desert locust swarms will invade China is of wide concern [9]. To answer this question, we first checked whether there are potentially suitable habitats for desert locusts within China, and then modeled the windborne movement of the desert locust to find out whether it is possible that locusts will reach China by crossing India and Myanmar.

## 2. Materials and Methods

### 2.1. Identifying Potentially Suitable Habitats for Desert Locusts within China

Previous studies showed that (i) the desert locust is well-adapted to live in arid and semi-arid habitats where annual precipitation is <400 mm [10], and at least 20 mm of rain falling in a short period (or its equivalent in run-off) is required for egg development [1]; (ii) the air temperature range for egg and hopper (nymph) development is between 20–35 °C [4]; and (iii) if there is more than seven days at 10 °C, egg mortality is considerably increased [11]. Therefore, areas with an annual precipitation ≤400 mm and air temperature in July ≥20 °C were identified as potentially suitable habitats for the desert locust in this study, and for the year-round breeding areas, the temperature in winter should be ≥10 °C. To do this, we retrieved the mean annual precipitation for 2000–2019 from the Climate Prediction Center Merged Analysis of Precipitation data, and the mean monthly air temperature for 2000–2019 from the National Center for Environmental Prediction (NCEP)/National Center for Atmospheric Research (NCAR) reanalysis data (https://www.ncep.noaa.gov/).

### 2.2. Modelling the Windborne Migration of Desert Locusts

#### 2.2.1. Swarm Movement at Low Altitudes

Gregarious swarms can move at low altitude (i.e., just a few meters above the ground) or, during strong convection, at high altitudes (hundreds of meters above ground) in daytime, while spending the night roosting in vegetation [2,4]. However, the swarms do not undertake the night-time long-distance flights of the solitarious phase [1]. Locust low-altitude fliers can stabilize their ground speed at about 4.0 m/s by varying their air speed and heading direction according to the wind speed, and thus they can move up to 150 km in one day by 10-hr sustained flying [2,4]. The number of days for desert locust spread into China was calculated by dividing the straight distance between the current occurrence area and China by the ground speed (4.0 m/s).

#### 2.2.2. Windborne Migration at High-Altitude

Desert locust swarms also can travel over hundreds and even thousands of kilometers by riding high-level winds, for example from North-West Africa to the British Isles in 1954, and from West Africa to the Caribbean, a distance of 5000 km, in about ten days in 1988 [12,13].

Firstly, we explored the wind characteristic at 850 hPa (about 1500 m above sea level) by analyzing the 2000–2019 historical climatic data derived from NCEP/NCAR reanalysis data, to find out whether there is a suitable high-speed wind stream to facilitate locust windborne migration into China. Then, we modelled the long-distance migration of the desert locust with the Hybrid Single-Particle Lagrangian Integrated Trajectory (HYSPLIT) model of the National Oceanic and Atmospheric Administration (NOAA) [14]. This model is designed for computing three-dimensional trajectories of air parcels and has been applied extensively to study migratory trajectories of many insect species [15,16]. Here, we made the following assumptions for the trajectory simulation of the desert locust: (i) high-flying swarming locusts do not have any single preferred orientation and turn back into the swarm when they reach the edges in order to maintain swarm cohesion, and thus swarm displacement will be the same as the wind speed and direction at the height of flight [1,2,12,13]; (ii) daytime flight occurs mostly from 09:00 h (local time, i.e., 03:00 UTC) to 19:00 h; (iii) locusts cannot fly at air temperatures below 20 °C; (iv) long-distance flight can be sustained for up to 10 days [12,13].

When this study began in late February 2020, the swarms in Pakistan and India were already mature and had laid eggs, and were thus less likely to spread over long distances [5,6]. Therefore, we focused on the next generation of adults in April–June. From the latest figures from FAO in May and early June, it was reported that new generation adults present in this area since April were moving eastward in June [7,17]. Consequently, our trajectory modelling and wind-field analysis concentrated on the April–June period.

## 3. Results

### 3.1. Potentially Suitable Habitat for the Desert Locust within China

The mean annual precipitation data showed that most areas in South Asia, Southeast Asia and East Asia are too wet for desert locust persistence (annual rainfall ≥400 mm), such as most of India, Myanmar and most of China (Figure 1). In China, only in parts of Xinjiang, Inner Mongolia, Tibet and Qinghai provinces, the annual rainfall is ≤400 mm. However, the mean air temperature in the Qinghai-Tibet Plateau is still quite cold in July (≤20 °C). Further, in most areas of China, the mean air temperature in January is ≤10 °C (Figure 1), and this shows that most areas of China are too cold for desert locusts to survive in winter. Taking this information together, only parts of Xinjiang and Inner Mongolia provinces could conceivably be suitable habitat for the desert locust in China, and these would just be ephemeral habitats in summer.

### 3.2. Modelling the Windborne Movement of the Desert Locust

#### 3.2.1. Low-Altitude Movement

Although we found only a few suitable habitats for desert locusts in China, there might be still a short-term threat if the swarms from the border between Pakistan and India enter China. However, the Himalayan mountains exceed 7000 m, and form a natural barrier stopping all or almost all insect migration—this is particularly likely with desert locusts as the lowest air temperatures (in the absence of sun) at which sustained swarm flight begins is 23–24 °C, and even in continuous bright sunshine flight does not begin at air temperatures much below 17 °C [1]. Thus, the only possible route that desert locust swarms could enter China is by moving eastward into Yunnan province after crossing India and Myanmar. The straight distance from the Pakistan–India border to Yunnan province is about 3000 km. Swarms from this region would need 20 days’ sustained flying to cover this distance if they just keep flying at low altitudes (near ground level), and this is quite impossible.

#### 3.2.2. Wind Characteristics at the Level of 850 hPa

As previously mentioned, this study focused on the period April–June. It was found that the winds at 850 hPa level (about 1500 m above sea level) blew consistently toward the east during daytime in these three months (Rayleigh test; April: mean value 88°, *r* = 0.58, *p* < 0.0001, *n* = 600; May: 80°, *r* = 0.76. *p* < 0.0001, *n* = 620; June: 53°, *r* = 0.66. *p* < 0.0001, *n* = 600; the *r*-value is a measure of the clustering of the angular distribution from 0 to 1), but the wind speed is quite slow (April: 3.32 ± 0.07 m/s, *n* = 600; May: 4.23 ± 0.08 m/s, *n* = 6 20; June: 4.27 ± 0.08 m/s, *n* = 600) (Figure 2).

#### 3.2.3. Modelling Windborne Migration at High Altitude

In total, 455 forward trajectories starting from a point at the border between Pakistan and India (27° N, 70° E) were calculated for each day in April–June over the last 5 years (2015–2019) (Figure 3). Most trajectories went eastwards, and the endpoints were located east of the startpoint (Rayleigh test; April: mean value 90°, *r* = 0.91, *p* < 0.0001, *n* = 150; May: 95°, *r* = 0.94, *p* < 0.0001, *n* = 155; June: 57°, *r* = 0.74, *p* < 0.0001, *n* = 150). Due to the slowness of the winds, migration distances were quite short (April: 917.3 ± 16.9 km; May: 1116.1 ± 11.5 km; June: 702.3 ± 12.9 km), and most trajectories thus ended within India (Figure 3). All these results indicate that it is impossible for desert locusts to reach China by 10 days’ windborne migration.

## 4. Discussion

The distribution of desert locusts is very well known and, during plagues, it can cover the arid and semi-arid region from the Atlantic coast of West Africa to eastern India. The most easterly distribution during invasion periods reaches eastern India and Bangladesh but does not include any part of China [1,4]. The potentially suitable area identified in this study was based on an annual precipitation ≤400 mm and mean air temperature in January ≥10°, and this area is consistent with the desert locust ‘recession area’, that is, the area occupied when there are few, if any, swarms present [1,4,10]. In China, only parts of Xinjiang and Inner Mongolia provinces could provide an ephemeral habitat in summer, and these regions are far away from any other desert locust breeding areas. During April–June, the winds at a height of about 1500 m above sea level blow eastward, but the wind speed is quite slow. In our trajectory simulations, most trajectories with 10 days’ migration ended within India, and the furthest just reached eastern India, or close to the border between India and Myanmar. The most easterly point of our trajectories is almost the same as the eastern border of the invasion area of desert locusts described in previous studies [2,8]. We therefore conclude that it is highly improbable that significant swarming populations of the desert locust will invade China.

This study was conceived in February, and what it predicted has already happened [7,17,18,19,20]. According to the figures from the FAO, desert locusts bred in Pakistan and India during March and April, and some immature adult groups formed in Pakistan in April [17,18]. In May and June, an increasing number of immature adult groups and swarms formed in Pakistan and the border area between India and Pakistan, and these swarms moved east, reaching northern Indian states such as Madhya Pradesh, Chhattisgarh, Uttar Pradesh and Bihar [7,19]. Small groups of immature adults arrived in Nepal from adjacent areas of Uttar Pradesh during strong southerly winds, and some of these groups even reached the base of the Himalayan foothills in late June [19]. In July, swarms prevailed in the northern states of Madhya Pradesh and Uttar Pradesh but then returned west with the onset of the monsoon to Rajasthan [20]. These swarms quickly matured and were seen copulating [20]. In summary, most swarms of desert locust from the border between Pakistan and India spread to the east in May and June, and just reached northern India and a few arrived in Nepal, but none were found in China [7,17,18,19,20]. This situation coincided with our results.

In the past, only one individual of the solitaria form of desert locust was detected in Zhangmu District, Nyalam County in Tibet (location at about 28.33° N, 86° E, elevation; 2250 m) on 29 April 1974, and this place is near the Nepali border on the southern slopes of the Himalayan Mountains [21]. It is obvious that this individual came from Nepal on strong southerly winds. Similarly, small groups of desert locusts also arrived in Nepal this June [19]. However, these particular groups caused little damage and quickly perished because the temperatures were too low and environmental conditions were not suitable for them.

We conclude that the desert locust will not invade China from Indian/Pakistan. In contrast, another invasive pest, the fall armyworm moth successfully entered China from India in late 2018 [8,22]. There are two probable explanations for the invasion process of the fall armyworm. Firstly, it can exist in a wide range of environmental conditions; it can breed year-round in most areas of Africa, Southwest Asia, South Asia, Indochina Peninsula and southern China [23], and the lands of these areas are connected or adjacent to each other. Therefore, it has extended its distribution generation by generation until it reached all of its potential breeding area [8]. Secondly, winds in northern India and Bangladesh are quite slow and would not facilitate fall armyworm eastward migration over large distances, but there is a very strong airstream between southern India and Myanmar in the rainy season in June–October (Figure 2c) [24], which would carry insects from southern India into Myanmar by crossing the Bay of Bengal. Our preliminary results from trajectory analyses showed that fall armyworm moths can cross the Bay of Bengal by a ≤36-h flight in September (unpublished data, G. Hu from Nanjing Agricultural University). Moreover, the fall armyworm in Myanmar was first detected on maize plants in the Mandalay and Ayeyawaddy Regions in late 2018 or early 2019, and these regions are on, or close to, the coast [25], thus reinforcing the view that fall armyworm migrate into Myanmar from India by crossing the Bay of Bengal. Recent studies have reported that there are frequent genetic exchanges in migratory insects between southern India and southern China, such as the brown planthopper (*Nilaparvata lugens*) [26] and tobacco cutworm (*Spodoptera litura*) [27], and this also suggests that insects can migrate from southern India into southern China. It should be emphasized that southern India is suitable for these insect species. By contrast, a large area between the China/Myanmar and Indian/Pakistan borders (most areas of India and Bangladesh) are not suitable for desert locusts to build and maintain their population (Figure 1). Desert locusts only invade these areas occasionally during plague years, but the populations soon decline or return back to the west. Because of this, locusts cannot invade China via the Myanmar route.

In this study, the migration of desert locusts was simulated with wind data, and our trajectories showed the longest distance that the desert locust can cover under ideal conditions; we also showed that the actual distance of eastward migration was much shorter. Our study was based on historical climate data (2000–2019), but the environmental conditions in China and adjacent countries may alter with climate change. However, the climate is changing gradually, and the ranges of terrestrial animals are shifting ~6–17 km per decade [28]; thus it is not possible that potential areas of habitat, suitable for desert locusts, will appear in China in the next one or two decades. Therefore, our conclusion that there is a very low risk of the desert locust invading China is reliable.

## 5. Conclusions

In China, only parts of Xinjiang and Inner Mongolia provinces can provide ephemeral habitat in summer, and these regions are far away from any other desert locust breeding area. During April-June, winds at the height of locust swarm flight blew eastward, but the wind speeds were quite slow and would not facilitate desert locust eastward migration over large distances. In summary, it is very unlikely that agriculturally significant populations of the desert locust will invade China. Up to late August 2020, desert locust individuals moving east just appeared in northern India and a few in Nepal but none were found in China, and this coincided with the results of our trajectory simulations.

## Figures and Tables

**Figure 1 insects-11-00628-f001:**
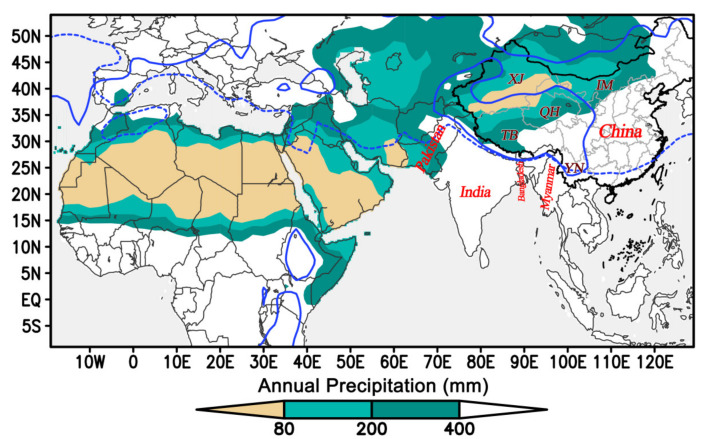
Mean annual precipitation (filled color), 10 °C isotherm in January (blue dotted line, the south side is ≥10 °C) and 20 °C isotherm in July (blue solid line, the south side is ≥20 °C) averaged for 20 years (2000–2019). Potentially suitable habitat for the desert locust should, at least, satisfy conditions with an annual precipitation ≤400 mm and air temperature in July ≥20 °C. As eggs cannot survive when the temperature is below 10 °C, for year-round breeding, the temperatures in winter should be above 10 °C. Taking all these things together, only small areas in Xinjiang and Inner Mongolia provinces could provide an ephemeral (summer) habitat for the desert locust in China. TB-Tibet, YN-Yunnan, QH-Qinghai, IM-Inner Mongolia.

**Figure 2 insects-11-00628-f002:**
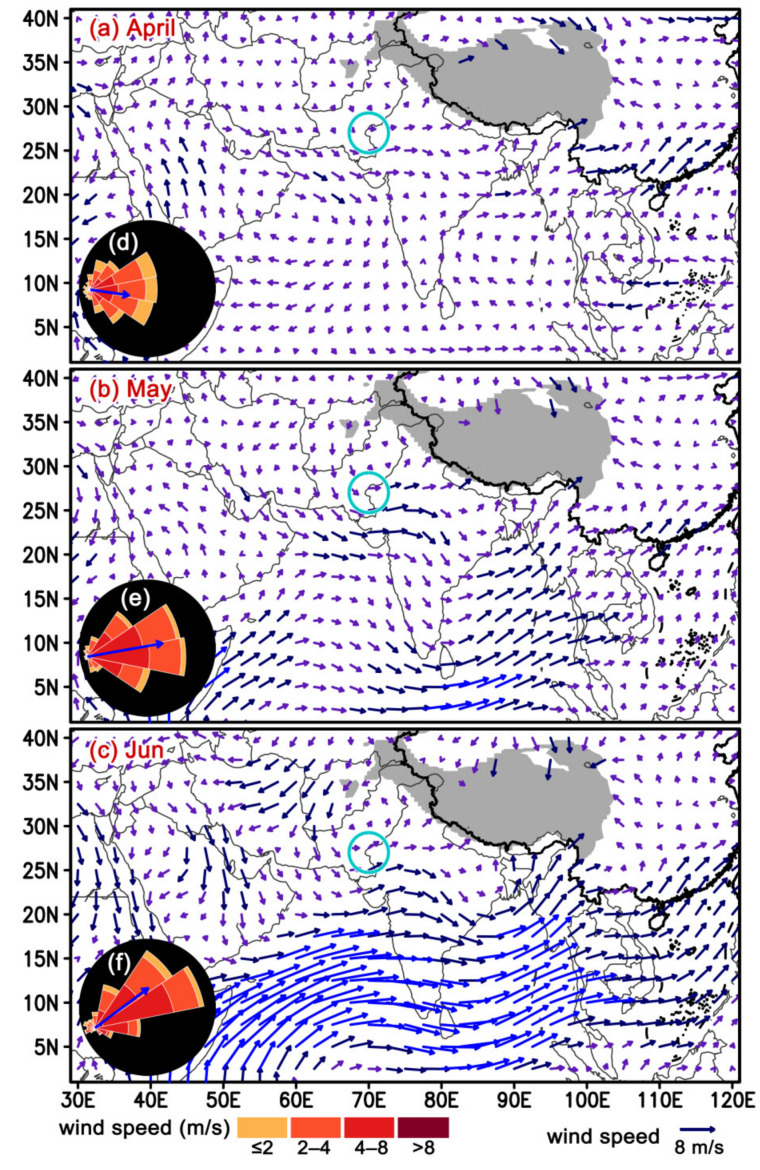
Mean wind (vector arrows) in (**a**) April, (**b**) May, (**c**) June, in the last 20 years (2000–2019). Subplots in (**d**–**f**) show the histogram distribution of downwind direction and wind speed in the border between Pakistan and India (indicated by blue circles). The grey filled area shows the Qinghai-Tibet Plateau.

**Figure 3 insects-11-00628-f003:**
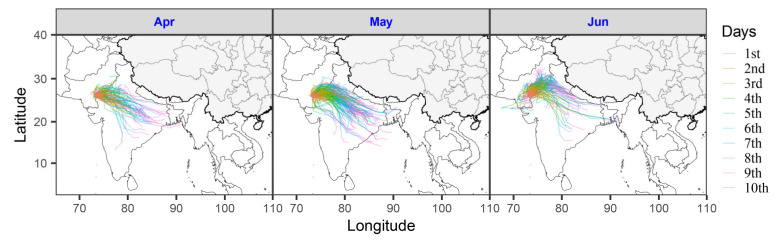
Simulated forward trajectories started from a point at the border between Pakistan and India (27° N, 70° E) in last 5 years (2015–2019). These trajectories were calculated by the Hybrid Single-Particle Lagrangian Integrated Trajectory (HYSPLIT) model.

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
