# Peer review of "Plagues of Desert Locusts: Very Low Invasion Risk to China"

_insects, 2020, doi:10.3390/insects11090628_

Round 1

Reviewer 1 Report

This is a mostly well written paper investigating the possibility of Desert Locusts invading China. To identify if there was suitable habitat in China for the locusts, the authors used various climatic conditions; minimum temperatures > 10oC in winter and above 20oC in summer (July) combined with an annual rainfall of below 400 mm. They then modelled the potential of locusts to invade these areas from Africa based on wind speed and direction, and the flight capability of the locusts. They concluded that it was unlikely the locusts would be able to invade China from the Indian/Pakistan border where swarms had laid eggs. There are some typos, misspelled words, and I think some headings the authors used when writing (ln180-181) that need addressing.

The authors used abiotic conditions to assess if there were suitable areas of habitat in China but this was rather crude and I think it requires further refined by looking at the vegetation present in these areas, and the timing of rainfall is what determines whether or not locusts will have suitable vegetation to enable growth, development and reproduction. The rationale for this work was that the armyworm had invaded China from India and Myanmar very recently and the concern was the Desert Locust could do the same thing. While the results presented here show that it is unlikely that locusts can migrate to China from Pakistan, what would be interesting is to know what is required for locusts to get to China? What enabled the armyworm, another pest that can migrate long distances to invade China?

While this manuscript is quite succinct, I feel it would benefit from some inclusion of some finer-scale detail and a re-focussing of the question. The authors’ state historically only one nymph has ever been seen in China (Ln171) and in its current form this paper would seem to be showing why this is. In its current form this paper would be better suited to a local applied agriculture journal.

Specific comments:

Ln53 – do you have a better reference as locusts cannot survive on 0 mm rainfall?

Ln64 typos in heading “modelling, windborne, migration and Desert” incorrectly spelled

Fig. 1. the temperature isotherms are unclear. Which side of the line is below 10 or 20oC and which below?

Ln 170 change “area” to “areas”

Ln180-181 remove

Author Response

There are some typos, misspelled words, and I think some headings the authors used when writing (ln180-181) that need addressing.

>>> All typos were corrected, and the sentence at L180-181was removed.

The authors used abiotic conditions to assess if there were suitable areas of habitat in China but this was rather crude and I think it requires further refined by looking at the vegetation present in these areas, and the timing of rainfall is what determines whether or not locusts will have suitable vegetation to enable growth, development and reproduction.

>>> We accept that our approach is “rather crude”.  However, the distribution of the desert locust is very well known, and the potentially suitable area identified in this study is consistent with the desert locust ‘recession area’. Thus our result is clear and reliable, and so we feel justified in making the conclusion that the majority of China is not suitable for the desert locust.

The rationale for this work was that the armyworm had invaded China from India and Myanmar very recently and the concern was the Desert Locust could do the same thing. While the results presented here show that it is unlikely that locusts can migrate to China from Pakistan, what would be interesting is to know what is required for locusts to get to China? What enabled the armyworm, another pest that can migrate long distances to invade China?

>>> The probable explanation for the invasion of China by the fall armyworm was added the manuscript:

(L197-220) : “Here we concluded that desert locust will not invade China from Indian/Pakistan. In contrast, another invasive pest, the fall armyworm moth successfully entered China from India in late 2018 [8,22]. There are two probable explanations for the invasion process of fall armyworm. Firstly, fall armyworm can exist in a wide range of environmental conditions; it can breed year-round in most area of Africa, Southwest Asia, South Asia, Indochina Peninsula and southern China, and the land of these areas are connected or adjacent [23]. Therefore, fall armyworm has extend its distribution generation by generation until it reached all of its potential breeding area [8]. Secondly, winds in the northern India and Bangladesh are quite slow and would not facilitate fall armyworm eastward migration over large distances, but there is very strong airstream between southern India and Myanmar in the rainy season in June-October (Fig. 2c) [24], which would carry insects from southern India into Myanmar by crossing the Bay of Bengal. Our preliminary results from trajectory analyses showed that fall armyworm moths can cross the Bay of Bengal by a ≤36 hours’ flight in September (unpublished data, G. Hu from Nanjing Agricultural University). Moreover, the fall armyworm in Myanmar was first detected on maize plants in the Mandalay Region and Ayeyawaddy Regions in late 2018 or early 2019, and these regions are on, or close to, the coast [25], thus reinforcing the view that fall armyworm migrate into Myanmar from India by crossing the Bay of Bengal. Recent researches reported that there are frequent genetic exchanges in migratory insects between southern India and southern China, such as brown planthopper (Nilaparvata lugens) [26] and tobacco cutworm (Spodoptera litura) [27], and this also suggested that insects would migrate from southern India into southern China. It should be emphasized that southern India is suitable for these insect species. By contrast, a large area between China/Myanmar and Indian/Pakistan border (most areas of India and Bangladesh) are not suitable for desert locust to build and maintain its population (Fig. 1). Desert locusts just invade these areas occasionally during plague years, but the populations soon decline or return back to the west. Because of this, desert locust cannot invade China via the Myanmar route.”

While this manuscript is quite succinct, I feel it would benefit from some inclusion of some finer-scale detail and a re-focussing of the question. The authors’ state historically only one nymph has ever been seen in China (Ln171) and in its current form this paper would seem to be showing why this is.

>>> The explanation of this was added (L190-L196).

 “In the past, only one individual of the solitaria form of desert locust was detected in Zhangmu District, Nyalam County in Tibet (location at about 28.33°N, 86°E, elevation; 2250 m) on 29 April 1974, and this place is near the Nepali border on the southern slope of the Himalayan Mountains [21]. It is obvious that this individual came from Nepal on strong southerly winds. Similarly, small groups of desert locust also arrived in Nepal in this June [19]. However, these particular groups caused little damage and quickly perished because the temperatures were too low and environmental conditions were not suitable for the desert locust.”

  Specific comments:

Ln53 – do you have a better reference as locusts cannot survive on 0 mm rainfall?

>>> We have changed this sentence to read, “…where annual precipitation is <400 mm”.

Ln64 typos in heading “modelling, windborne, migration and Desert” incorrectly spelled

Ln 170 change “area” to “areas”

Ln180-181 remove

>>> All typos were corrected, and the sentence at Ln180-180 was removed.

Fig. 1. the temperature isotherms are unclear. Which side of the line is below 10 or 20oC and which below?

>>> “south side is ≥ 10°C” & “south side is ≥ 20°C” were added in the caption of Fig. 1.

Reviewer 2 Report

This manuscript is an interesting commentary on the likelihood of the desert locust plagues in China based on ecological modeling. Apparently, this study was conceived in February, and it predicts what will happen in April-June 2020, based on mean annual precipitation, mean monthly air temperature, and particle trajectory model. Now, it is July 2020 when this review is made, and we have actual data regarding the desert locust migration patterns in April to June 2020 based on FAO data. In a way, we can validate the model’s projection.

This study identifies that there are few suitable habitats for the desert locust in China, and that if the swam ever reaches China, it would be through the Yunnan Province. This is a logical conclusion based on the fact that a more direct route from India to China has a major geological barrier – the Himalayas. In June 2020, the desert locust swarm reached Kathmandu, Nepal. This particular swarm caused some damages, but quickly perished because the temperature of too low and environmental conditions were not suitable for the desert locust.

Interestingly, the simulated forward trajectories shown in Figure 3 predict southeastern trajectories. In reality, what happened was that the desert locust swarms reached Nepal, and some headed west back to Pakistan. Predicting locust swarms requires not only abiotic conditions, but also phase status, local density, vegetation patterns, and developmental timing and with a limited number of parameters used in the model, the accuracy is compromised.

However, the general premise that China is probably not going to be affected by the desert locust is probably true. This is of course based on the assumption that the climate patterns will be stable. With Climate change, the environmental conditions in China and adjacent countries may change, which might promote potential invasion in the future.

This study is not really a rigorous scientific analysis, but more of a prediction and a commentary.

Author Response

Now, it is July 2020 when this review is made, and we have actual data regarding the desert locust migration patterns in April to June 2020 based on FAO data. In a way, we can validate the model’s projection.

>>> A long paragraph was added in the discussion on this point (L177-L189).

Interestingly, the simulated forward trajectories shown in Figure 3 predict southeastern trajectories. In reality, what happened was that the desert locust swarms reached Nepal, and some headed west back to Pakistan. Predicting locust swarms requires not only abiotic conditions, but also phase status, local density, vegetation patterns, and developmental timing and with a limited number of parameters used in the model, the accuracy is compromised. However, the general premise that China is probably not going to be affected by the desert locust is probably true. This is of course based on the assumption that the climate patterns will be stable. With Climate change, the environmental conditions in China and adjacent countries may change, which might promote potential invasion in the future.

>>>We agree with the general premise of the referee, however we stand by our main result and general conclusion, which is summarized by the concluding statement of our paper, repeated here:

(L221-L228): “the migration of desert locust was simulated with wind data, and our trajectories showed the longest distance that the desert locust can cover under ideal conditions; we also showed that the actual distance of eastward migration was much shorter. The climate is changing gradually, and the ranges of terrestrial animals are shifting ~6 -17 kilometers per decade [28]; thus it is not possible that potential areas of habitat, suitable for desert locusts, will appear in China in the next one or two decades. Therefore, our conclusion that there is a very low risk of desert locust invading China is reliable.”